# MI-Grad-CAM: Letting Your Model Reveal What's Most Informative

## Abstract

With the growing role of machine vision in critical applications such as healthcare, achieving precise and interpretable decision-making is crucial. Class Activation Mapping (CAM) is widely used for visual explanations in computer vision, but improving its interpretability remains an open research area. In this work, we introduce MI-Grad-CAM, a novel post-hoc visual explanation method that provides clearer, causally-driven insights into how CNNs reach their conclusions by prioritizing causality over mere correlation. MI-Grad-CAM generates class-specific visualizations by weighting feature maps based on normalized mutual information between the input image and feature maps, combined with gradient information of the predicted class with respect to these feature maps. This approach strengthens the causal link between explanations and model predictions, supported by counterfactual analysis to verify causality. We also propose the Harmonized Confidence Index (HCI), a new evaluation metric to measure explanation effectiveness. Our method demonstrates robust performance in both qualitative and quantitative evaluations, achieving competitive or superior results compared to state-of-the-art methods, particularly in terms of explanation faithfulness and model reliability. The source code for our proposed method is available at
`https://anonymous.4open.science/r/MI-Grad-CAM-C206`.

## 1 Introduction

With the increasing use of deep learning in vision-based applications such as healthcare automation Helaly et al. (2024); Huang et al. (2024); Cheng et al. (2024), self-driving vehicles Hao et al. (2024); Tonderski et al. (2024), and 3D data analytics El Banani et al. (2024); Varma et al. (2024); Kollias et al. (2024), ensuring transparency and reliability in these systems has become essential. Convolutional neural networks (CNNs) and encoder-decoder architectures often demonstrate exceptional performance; however, they are frequently challenging to interpret due to their complex structures and "black box" nature Lipton (2018); Liang et al. (2021); Arrieta et al. (2020). This lack of interpretability makes it difficult for end-users to understand the basis of model predictions, particularly in high-stakes applications. Consequently, there is a critical need for explainable artificial intelligence (XAI) methods that can make model inferences more accessible and interpretable, especially for tasks involving visual data Ali et al. (2023).

Current XAI approaches for visual interpretability include gradient-based Wang et al. (2024), perturbation-based Ribeiro et al. (2016), decomposition-based Gu et al. (2019); Iwana et al. (2019); Montavon et al. (2017), and CAM-based methods Muhammad & Yeasin (2020); Omeiza et al. (2019); Smilkov et al. (2017); Karmani et al. (2024). CAM-based methods are particularly popular, as they highlight important regions within input images by generating a weighted combination of feature maps. The foundational CAM approach Zhou et al. (2016) provided class-specific explanations by linearly combining feature maps from the final convolutional layer. However, this method required a global average pooling (GAP) layer, which limited its adaptability across various architectures. Grad-CAM was subsequently developed to address this limitation by utilizing the gradients of the predicted class relative to feature maps, enhancing flexibility across model architectures Selvaraju et al. (2017).

Further advancements, including Grad-CAM++ Chattopadhay et al. (2018), Integrated Grad-CAM Sattarzadeh et al. (2021), and XGrad-CAM Fu et al. (2020), improved Grad-CAM's ability to cap-

ture finer details in visual explanations. However, these methods still encountered challenges, such as the "shattered gradients" issue, which can hinder interpretability. To address these limitations, decomposition-based methods like Score-CAM Wang et al. (2020) were introduced, offering a gradient-free approach that leverages class scores from perturbed images to weigh feature maps. Variants like Score-CAM++ Chen & Zhong (2022) and Integrated Score-CAM Naidu et al. (2020) further refined this technique, while Eigen-CAM incorporated a principal component analysis method for weight calculation Muhammad & Yeasin (2020).

Despite these advances, CAM-based methods still exhibit certain limitations: they often lack a direct relationship between feature maps and the input data, and they do not produce reliable causal explanations Adebayo et al. (2018); Kindermans et al. (2019); Hooker et al. (2019). These shortcomings highlight the need for methods that extend beyond correlation-based insights, offering explanations grounded in causal relationships. Further, existing evaluation metrics for measuring explanation effectiveness remain limited. This underscores importance of developing robust explanation techniques and establishing reliable metrics to evaluate quality and reliability of explanations effectively.

### 1.1 Contributions

Overseeing the limitations of the existing CAM-based visualization methods, we introduce MI-Grad-CAM, a new approach for explaining model predictions. MI-Grad-CAM uses mutual information between feature maps and the input image, along with gradients of the predicted class with respect to the feature maps, to assign weights to the activation maps, generating causal visualizations. Our main contributions are summarized as follows:

- We propose a novel informed CNN visual explanation method named MI-Grad-CAM, bringing information theory and CAM-based explanations closer.
- We demonstrate the causality of the generated saliency maps through counterfactual analysis. MI-Grad-CAM shows notable performance in delivering causal representations and capturing latent properties essential for effective CAM-based visual explanations.
- We evaluate the fairness and effectiveness of the generated class activation maps from both quantitative and qualitative perspectives using Average Drop (AD) / Average Increase (AI), Deletion/Insertion Curve metrics, and visual inspection.
- We introduce HCI, a balanced metric that combines AI and AD scores to provide a single, interpretable measure of the quality of the explanation.

## 2 Background

Let $Y = \mathcal{F}(X)$ represent a model, where an input $X$ produces a probability distribution $Y$. For a given layer $l$; $A_l$ and ${A_l}^k$ represent the activation maps of the $l$-th layer and its $k$-th channel map.

### 2.1 Mutual Information (MI)

MI is a dependency metric that quantifies how much information one random variable contains about another. Let $V_1$ and $V_2$ be two random variables with corresponding entropies $H(V_1)$ and $H(V_2)$, defined as:

$$H(V_1) = -\sum_{v_1} P(v_1) \log P(v_1), \quad H(V_2) = -\sum_{v_2} P(v_2) \log P(v_2), \tag{1}$$

$$H(V_1, V_2) = -\sum_{v_1} \sum_{v_2} P(v_1, v_2) \log P(v_1, v_2). \tag{2}$$

where $P(v_1)$ and $P(v_2)$ are the marginal probabilities of $V_1 = v_1$ and $V_2 = v_2$ respectively, $P(v_1, v_2)$ is their joint probability and $H(V_1, V_2)$ is the joint entropy of $V_1$ and $V_2$. The MI between the two random variables is defined as:

$$I(V_1; V_2) = H(V_1) + H(V_2) - H(V_1, V_2) \tag{3}$$

$$= \sum_{v_1} \sum_{v_2} P(v_1, v_2) \log \left( \frac{P(v_1) P(v_2)}{P(v_1, v_2)} \right) \tag{4}$$

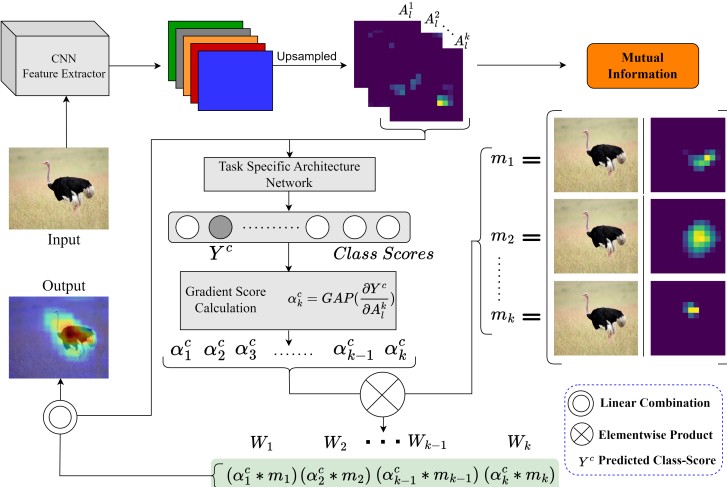

Figure 1: Pipeline of MI-Grad-CAM. Activation maps from a CNN are weighted using MI with the input and pooled class gradients, then linearly combined to generate the final explanation.

## 2.2 CLASS ACTIVATION MAPPING (CAM)

CAM is an explainable technique that highlights important regions by calculating a linearly weighted combination of activations in the last convolutional layer before the GAP layer. However, CAM requires that feature maps appear before the softmax layer, limiting its applicability to specific network architectures.

**Definition 1** *Let a CNN model $\mathcal{F}$ include a preceding convolutional layer $l - 1$, a GAP layer $l$ followed by a fully connected layer $l + 1$. For a given class of interest c, vanilla CAM can be represented as*

$$L_{CAM}^c = ReLU \left( \sum_k \alpha_k^c A_{l-1}^k \right) \tag{5}$$

*where $\alpha_k^c = w_{l,l+1}^c[k]$, with $w_{l,l+1}^c[k]$ representing the weight between layers l and l+1 for the k-th feature map. This method faces generalizability issues across different model architectures, as it is only applicable to architectures where the final convolutional layer is followed by a GAP layer and a fully connected layer, without requiring re-training on altered architectures. The CAM has some limitations discussed in the appendix.*

## 2.3 CAUSAL EXPLANATIONS

In neural networks, the output is derived from the input through a sequence of layer-wise transformations. Each layer generates $A_l^k$, which act as intermediate causes contributing to the final output.

**Definition 2** *A model can be represented as a composition of functions $Y = \mathcal{F}(X) = f_L(f_{L-1}(...f_2(f_1(X))...))$. The feature map at layer l is $A_l^k = f_l^{(k)}(f_{l-1}(...f_2(f_1(X))...))$. The causal relationship is expressed as $P(A_l^k, Y) = P(Y \mid A_l^k) \cdot P(A_l^k)$, indicating that Y depends on $A_l^k$. The causal effect estimate can thus be denoted using do-calculus Zhang (2007) as $P(Y \mid do(A_l^k = a))$.*

## 2.4 CONDITIONAL INDEPENDENCE IN CNNS

The CNN model represented as $Y = f(X)$, where $l-1$, $l$, and $l+1$ are consecutive layers producing activations $A_{l-1}^k$, $A_l^k$, and $A_{l+1}^k$ respectively. The activation $A_{l-1}^k$ captures all relevant information from the input $X$ needed to determine $A_l^k$. This dependency is expressed by the conditional independence statement $(P(A_l^k \mid A_{l-1}^k, X) = P(A_l^k \mid A_{l-1}^k)$ indicating that $A_{l-1}^k$ d-separates $X$ from $A_l^k$ Fritz & Klingler (2023). This principle extends across all layers of the CNN, implying that

---

**Algorithm 1:** MI-Grad-CAM Algorithm

---

**Input:** Input Image $X$, CNN model $\mathcal{F}(.)$, layer $l$, number of channels $K$, predicted class score $Y^c$
**Output:** $L^c_{MI-Grad\ CAM}$

**1** $A_l \leftarrow f_l(X)$                                 `/* Get activation of layer l */`
**2** $Y^c \leftarrow head\ network(\mathcal{F}(X))$                    `/* Compute class scores */`
**3** **for** $k = 1$ *to* $K$ **do**
**4**     $\alpha^c_k \leftarrow GAP(\frac{\partial Y^c}{\partial A^k_l})$                  `/* Calculate pooled gradients */`
**5**     $A^k_l \leftarrow Up(A^k_l),\ X \leftarrow Gr(X)$         `/* Upsample `$A^k_l$` and grayscale X */`
**6**     $A^k_l \leftarrow flat(A^k_l),\ X' \leftarrow flat(X)$     `/* Flatten activation map and input */`
**7**     $m_k \leftarrow I(A^k_l; X')$                 `/* Calculate mutual information */`
**8** **end**
**9** $m_k \leftarrow m_k/(\sum^K_{k=1} m_k)$          `/* Get normalized mutual information */`
**10** **for** $k = 1$ *to* $K$ **do**
**11**     $w^c_k \leftarrow \alpha^c_k * m_k$             `/* Compute weighted feature maps */`
**12** **end**
**13** $L^c_{MI-Grad\ CAM} \leftarrow ReLU(\sum_k w^c_k A^k_l)$

---

the information conveyed by the activations in the final layer encapsulates the information from all preceding layers.

## 3 MI-GRAD-CAM: PROPOSED APPROACH

The working of MI-Grad-CAM is illustrated in Fig. 1, which combines MI between activation maps and the input with the gradient information of the predicted class score relative to each activation map to determine importance metrics. Unlike methods that rely exclusively on the predicted class to assign importance to each activation map and explain the model's decision-making process, our approach integrates these two sources of information to achieve a more comprehensive weighting.

**Definition 3** *Let for given input $X$, the CNN model $Y = \mathcal{F}(X)$, and the activation map $A^k_l$ for $k^{th}$ channel at layer $l$. The probabilistic estimates of activation $P(A^k_l)$ and input $P(X)$ are defined as:*

$$P(A^k_l) = P(\ flat(\ Up(\ A^k_l))) \tag{6}$$
$$P(X) = P(\ flat(\ Gr(\ X))) \tag{7}$$

*where $Up(.)$ up-samples $A^k_l$ to the input size, $flat(.)$ performs a flattening operation that maps the upsampled activation map from a 2D space to a 1D vector, and $Gr(.)$ denotes the grayscale transformation applied to the input. The joint probability between $A^k_l$ and $X$ is represented as $P(A^k_l, X)$. The MI measure $I(A^k_l; X)$ between $A^k_l$ and $X$ is calculated as:*

$$I(A^k_l; X) = \sum_{a \in A^k_l} \sum_{x \in X} P(A^k_l = a, X = x) \log \left( \frac{P(A^k_l = a, X = x)}{P(A^k_l = a)P(X = x)} \right) \tag{8}$$

The normalized MI is then used as input information-centric weighting component:

$$m_k = \frac{I(A^k_l; X)}{\sum^N_{k=1} I(A^k_l; X)} \tag{9}$$

For the predicted class $c$, the gradient information-based weight component of each activation map is calculated to incorporate class-specific information as:

$$\alpha^c_k = \text{GAP} \left( \frac{\partial Y^c}{\partial A^k_l} \right), \quad \text{where GAP(.) denotes global average pooling.} \tag{10}$$

The overall weighting values of each activation map can then be expressed as:

$$w^c_k = m_k \cdot \alpha^c_k, \quad \text{where } (\cdot) \text{ denotes scalar multiplication.} \tag{11}$$

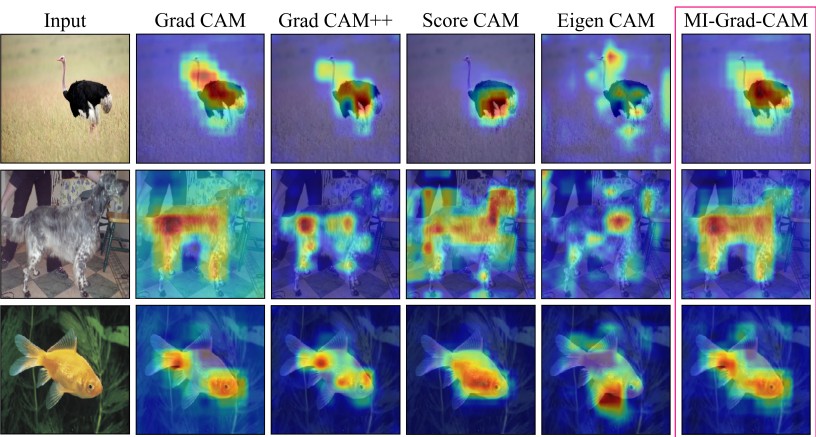

Figure 2: Visualization of state-of-the-art CAM-based explanation methods with MI-Grad-CAM

Finally, we define our proposed visual explanation method, MI-Grad-CAM, in Definition 4 and detailed steps are provided in Algorithm 1, where *head network(.)* represents the head architecture added on top of the CNN to obtain task-specific outputs.

**Definition 4** *MI-Grad-CAM: Given a convolutional layer l of a model $\mathcal{F}$, and an input $X$, we define MI-Grad-CAM ($L_{MI-Grad-CAM}$) as follows:*

$$L^c_{MI\text{-}Grad\text{-}CAM} = \text{ReLU}\left(\sum_k w^c_k A^k_l\right), \quad \text{where } w^c_k \text{ is defined in Eq. 11.} \quad (12)$$

MI-Grad-CAM captures the feature-input relationship by quantifying MI and reduces class-specific biases, providing causal explanations that account for both linear and non-linear dependencies. Although the last convolutional layer is the primary focus, it encapsulates all preceding information as explained in **Def.** 2; any intermediate convolutional layer can also serve as a target within our proposed framework. A mathematical overview and proof of our proposal is provided in the Appendix.

## 4 EXPERIMENTS

First, we conduct a qualitative evaluation using visualizations on ImageNet validation dataset. This is followed by a faithfulness assessment of our method using Average Drop (AD), Average Increase (AI), and Deletion and Insertion Curve metrics. Finally, we assess the causality of our method through Counterfactual Analysis visualizations.

### 4.1 QUALITATIVE VISUAL INSPECTION

We present a comparative analysis of the heatmaps generated by our proposed MI-Grad-CAM alongside various state-of-the-art methods, including gradient-based, score-based, and eigen-based approaches, for qualitative evaluation. Our method excels in producing visually credible and clearer heatmaps, accurately highlighting the actual spatial regions being focused by the model with reduced random noise. The results, shown in Fig. 2, visually demonstrate that MI-Grad-CAM introduces less random noise compared to other CAM methods, including Grad-CAM, Grad-CAM++ Chattopadhay et al. (2018), Score-CAM Wang et al. (2020), and Eigen-CAM Muhammad & Yeasin (2020). Additional visualizations on ResNet-50 are provided in the **Appendix** A.10.

### 4.2 CLASS-SELECTIVE VISUALIZATION

We demonstrate the class-aware discriminative ability of MI-Grad-CAM in distinguishing between different classes, as shown in Fig. 3. Masked images for each class provide a qualitative analysis of the class sensitivity performance of our proposed method. The VGG-16 model attributes a confidence of 99.2% for the 'zebra' class and 0.025% for the 'African elephant' class when classifying the given input. Our method accurately highlights the relevant regions for both classes, despite the highly contrasting probabilities between the two.

Table 1: Quantitative evaluation results for AD, AI, HCI, and win % with thresholds of 0.5 and 0.7.

| M | Method | 0.5 | | | | 0.7 | | | |
|---|---|---|---|---|---|---|---|---|---|
| | | (AD) ↓ | (AI) ↑ | HCI ↑ | Win ↑ | (AD) ↓ | (AI) ↑ | HCI ↑ | Win ↑ |
| VGG-16 | Grad CAM | 93.978 | 39.265 | 0.104 | 13.3 | 97.221 | 21.000 | 0.049 | 16.6 |
| | Grad CAM++ | 97.925 | 16.133 | 0.037 | 6.6 | 98.885 | 14.057 | 0.020 | 6.6 |
| | Score CAM | 87.015 | 40.882 | 0.196 | 23.3 | 94.590 | 27.345 | 0.090 | 26.6 |
| | XGrad CAM | 96.932 | 39.720 | 0.072 | 3.3 | 98.501 | 38.139 | 0.028 | 3.3 |
| | Eigen CAM | 98.729 | 14.676 | 0.035 | 13.3 | 99.211 | 13.175 | 0.014 | 6.6 |
| | MI-Grad-CAM | **83.999** | **42.565** | **0.243** | **40.0** | **94.327** | 27.583 | **0.093** | **40.0** |
| ResNet-50 | Grad CAM | 89.396 | 46.279 | 0.172 | 6.6 | 92.836 | 25.724 | 0.112 | 6.6 |
| | Grad CAM++ | 92.013 | 37.017 | 0.131 | 13.4 | 96.847 | 22.428 | 0.055 | 13.4 |
| | Score CAM | **52.960** | 22.044 | **0.300** | 32.3 | **56.797** | 18.780 | **0.261** | 32.3 |
| | XGrad CAM | 95.930 | 32.770 | 0.086 | 10.0 | 97.812 | 27.876 | 0.040 | 10.0 |
| | Eigen CAM | 95.649 | 27.471 | 0.084 | 3.3 | 98.206 | 19.299 | 0.032 | 3.3 |
| | MI-Grad-CAM | 83.419 | **50.279** | 0.249 | **34.3** | 90.437 | **34.724** | 0.150 | **34.3** |
| DenseNet-121 | Grad CAM | 76. 939 | 46.648 | 0.308 | 16.6 | 83.655 | 37.049 | 0.226 | 13.4 |
| | Grad CAM++ | 80.500 | 31.567 | 0.241 | 23.3 | 89.918 | 17.550 | 0.128 | 26.7 |
| | Score CAM | 73.738 | 34.316 | 0.297 | 13.3 | **79.637** | 28.009 | 0.235 | 16.7 |
| | XGrad CAM | 87.462 | 41.302 | 0.192 | 13.3 | 90.664 | 37.054 | 0.149 | 10.0 |
| | Eigen CAM | 88.401 | 17.427 | 0.139 | 0.0 | 92.581 | 13.070 | 0.094 | 0.0 |
| | MI-Grad-CAM | **72.948** | **48.651** | **0.347** | **33.3** | 82.664 | **38.834** | **0.243** | **33.4** |

The masked images demonstrate excellent class sensitivity. As shown in Fig. 3, the African elephant is localized along with contextual information of its habitat, where it is commonly found in ImageNet images. This highlights the model's actual areas of focus in the decision-making process and reflects the benefit of incorporating MI into the weighting parameter. Given that each activation map's weight is causally linked to both MI and target class gradients, MI-Grad-CAM is well-equipped to differentiate between categories while capturing relevant contextual information, enhancing its class discriminative ability.

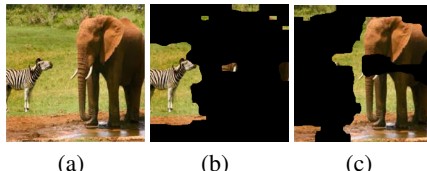

(a)  (b)  (c)

Figure 3: Class-based visualization: (b) and (c) are generated with respect to the classes 'zebra' and 'African elephant'.

### 4.3 MULTI-OBJECT VISUALIZATION

MI-Grad-CAM demonstrates effective localization not only on images with single objects or elements of interest but also on images containing multiple objects of the same class. The results in Fig. 4 demonstrate the performance of MI-Grad-CAM on multi-object images. Saliency maps generated by Eigen-CAM attributes importance only to one among the multiple objects present, whereas, Saliency maps generated by MI-Grad-CAM show better localization by identifying all the objects. MI-Grad-CAM also effectively captures informative contextual spatial regions that the model deems essential for its function.

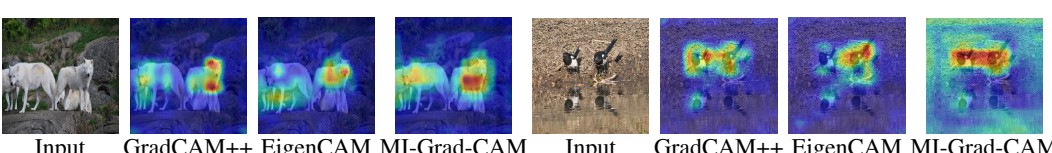

Input  GradCAM++  EigenCAM  MI-Grad-CAM  Input  GradCAM++  EigenCAM  MI-Grad-CAM

Figure 4: Multi-object visualization results. MI-Grad-CAM improves upon other methods by covering important spatial locations and contextual information, providing a holistic explanation of all objects in the image.

### 4.4 QUANTITATIVE FAITHFULNESS EVALUATION

We evaluate the faithfulness of MI-Grad-CAM in generating reliable explanations using the Average Drop (AD), Average Increase (AI), Area Under the Deletion/Insertion Curve, and the proposed Harmonized Confidence Index (HCI) metrics, as described below. Unless stated otherwise, quantitative analysis experiments use the final convolutional layer of the VGG-16 Simonyan & Zisserman (2014), ResNet-50 He et al. (2016), and DenseNet-121 Huang et al. (2017) models pre-trained on ImageNet.

**1. Average Drop (AD):** Following Chattopadhay et al. (2018), we use the AD metric to quantify the target score loss when perturbing the input image with a CAM mask. Unlike previous studies that apply pixel-wise masking with activation map pixels, we normalize and rank the activation map pixels intensity, **by setting a threshold**. We mute pixels in the original image corresponding to activation map pixels below this threshold to assess the importance of regions emphasized by our method. We calculate AD as $\sum_{i=1}^{N} \frac{\max(0, Y_i^c - O_i^c)}{Y_i^c} \times \frac{100}{N}$, where $Y_i^c$ is the predicted score for class $c$ on the original image, and $O_i^c$ is the predicted score on the CAM-masked image. Lower AD values indicate better performance.

**2. Average Increase (AI):** We use the AI metric to measure the target score gain when introducing the most valued CAM pixels into a baseline input, using a threshold value. Pixels in the input corresponding to activation map pixels above this threshold are added to the baseline input. We calculate AI as $\sum_{i=1}^{N} \frac{\max(0, O_i^c - B_i^c)}{O_i^c} \times \frac{100}{N}$, where $O_i^c$ and $B_i^c$ denote the predicted scores for class $c$ on input with CAM-focused pixels and baseline input, respectively. $N$ is the total number of input samples. Higher AI values indicate a more effective CAM approach.

**3. Deletion/Insertion Curve:** We plot Deletion and Insertion Curves by measuring prediction scores as a function of the fraction of pixels deleted or inserted, respectively. For the Deletion Curve, we rank important pixels identified by the CAM in descending order and remove them in stages, recording the prediction score at each step. The Area Under the Deletion Curve (AUC) quantifies faithfulness of the CAM method, where a lower AUC indicates greater robustness. For the Insertion Curve, we arranged important pixels in ascending order and add certain ratio of pixels in ascending order of importance to get the corresponding class scores. The higher AUC signifies a more robust CAM approach.

**4. Harmonized Confidence Index (HCI):** Since AD and AI measure confidence retention and enhancement independently, they may yield conflicting results across CAM approaches. To resolve this, we propose the HCI, an integrated metric that combines AD and AI into a balanced score by calculating the harmonic mean:

$$HCI = 2 \left( \frac{(1 - AD) \cdot AI}{(1 - AD) + AI} \right) \qquad (13)$$

Lower AD and higher AI values signify better performance, so a higher HCI score reflects a more interpretable approach. We provide a detailed explanation of HCI in the Appendix.

**5. Win Percentage:** Alongside AD, AI and HCI, we calculated the win percentage metric as in previous XAI studies Chattopadhay et al. (2018). The win percentage provides insight on the number of times the average drop of a particular approach was the least among the other approaches, i.e. it provides a per-sample analysis in place of the batch analysis provided by AD. This value, expressed in percentage, provides the win percentage metric.

#### 4.4.1 PERFORMANCE INSIGHTS

Table 1 summarizes the results over a subset of the ImageNet (ILSVRC2012) validation data with 2000 samples. MI-Grad-CAM achieves an AD of 83.9% and 72.9% and an AI of 42.5% and 48.6% on 0.5 threshold for VGG-16 and DenseNet-121, respectively, outperforming other CAM-based methods. Similarly, MI-Grad-CAM scores AD of 94.3% and 82.6% and AI of 27.5% and 38.8% on VGG-16 and DenseNet-121, respectively with a threshold of 0.7. MI-Grad-CAM demonstrated superior performance, achieving HCIs of 0.243 (VGG-16) and 0.347 (DenseNet-121) with a 0.5 threshold, and 0.093 (VGG-16) and 0.243 (DenseNet-121) with a 0.7 threshold. In addition to AD, AI and HCI MI-Grad-CAM also outperforms the other approaches in terms of the win percentage metric by attaining the highest win percentage in all scenarios.

To further evaluate the faithfulness of our approach, Table 2 presents the average performance of MI-Grad-CAM on VGG-16 for deletion and insertion curves, complementing the AD and AI results.

Table 2: Comparative analysis for deletion and insertion curves using average AUC on VGG-16.

| Method | Deletion | Insertion |
|---|---|---|
| Grad CAM | 0.36449 | 0.01047 |
| Grad CAM++ | 0.40690 | 0.00151 |
| Score CAM | 0.35470 | 0.01124 |
| XGrad CAM | 0.35909 | 0.00767 |
| Eigen CAM | 0.35603 | 0.00755 |
| MI-Grad-CAM | **0.34888** | **0.01217** |

Table 3: Quantitative localization results for the pointing game (hit-rate) and EBPG on VGG-16.

| Method | hit-rate | EBPG (%)↑ |
|---|---|---|
| Grad CAM | 0.60 | 53.714 |
| Grad CAM++ | 0.60 | 54.187 |
| Score CAM | 0.56 | 54.905 |
| XGrad CAM | 0.53 | 52.438 |
| Eigen CAM | **0.76** | 54.164 |
| MI-Grad-CAM | 0.60 | **55.258** |

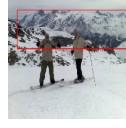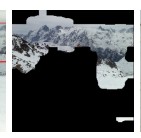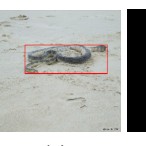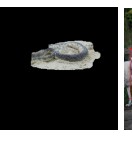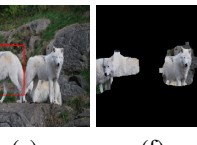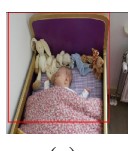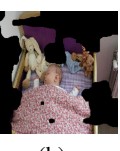

| (a) | (b) | (c) | (d) | (e) | (f) | (g) | (h) |

Figure 5: Visualization of localization results along with ground truths. (a), (c), (e), and (g) represent Alp, water snake, white wolf, and bassinet image, respectively, while (b), (d), (f), and (h) show the corresponding localizations.

Following the method outlined in Wang et al. (2020), we remove or insert pixels by setting their values to 0 or their original pixel intensity. At each step, 1% of the image pixels are deleted or inserted, continuing until 100 steps. The results in Table 2 highlight the superior performance of our method on both deletion and insertion curve metrics.

## 4.5 LOCALIZATION EVALUATION

The applicability of CAM methods in localization tasks, such as segmentation, unsupervised object detection, and self-erasing systems, makes localization evaluation an essential part of assessing the quality of generated saliency maps. Fig. 5 presented visualizations of localization results generated by MI-Grad-CAM on images from the ImageNet (ILSVRC2012) validation dataset using the pre-trained VGG-16 model. We conduct localization assessment by masking the image with MI-Grad-CAM to segment the most important objects and contextual information from the input.

We scale the activation maps to match the dimensions of the input image and exclude pixels with intensity values below 50% in the activation map for a given class. Further quantitative localization evaluation is performed using the pointing game and energy-based pointing game (EBPG) Wang et al. (2020) metrics. In the pointing game, each instance receives a 'hit' or a 'miss' depending on whether the maximum point or pixel in the saliency map falls inside or outside the ground truth bounding box annotations. EBPG provides a more detailed assessment by considering the energy of the saliency map within the target bounding box instead of just the maximum intensity point. Table 3 presents the pointing game and EBPG statistics for MI-Grad-CAM and other CAM methods.

Eigen CAM achieves the highest hit rate in the pointing game, with a score of 0.76, while MI-Grad-CAM performs consistently with other methods, achieving a hit rate of 0.6. Notably, MI-Grad-CAM demonstrates the best performance in terms of EBPG (%) for localization tasks, with an EBPG (%) score of 55.258%.

## 4.6 COUNTERFACTUAL ANALYSIS

As discussed in Hooker et al. (2019) and referenced in Sec 2.2, saliency maps should provide causal explanations to ensure robustness and capture both object and contextual information. Counterfactual examples can be generated by perturbing or setting specific pixels to defined values, as outlined in **Def.** 2. To ensure causal explanations, the feature map weights of both actual and counterfactual images should display variations. Fig. 6 shows counterfactual analysis plots, illustrating the feature map weights assigned in both actual and counterfactual setups for test samples.

The results are based on a small perturbation of '+0.1' applied randomly to pixel intensities to create counterfactual samples. Notably, this minor adjustment leads to significant variations in feature map weights between actual and counterfactual setups, particularly for the most important channels.

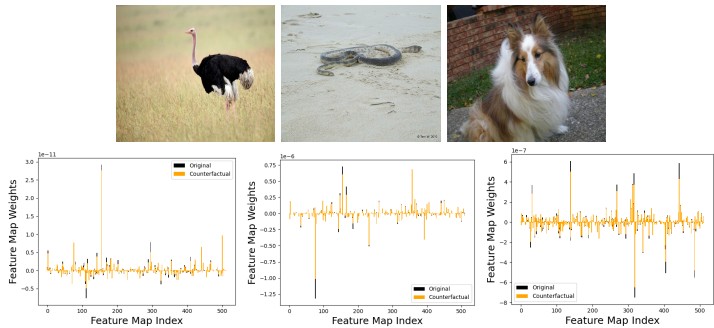

Figure 6: Counterfactual analysis results. Variations in the weights corresponding to feature maps for actual and counterfactual inputs indicate the causality of explanations.

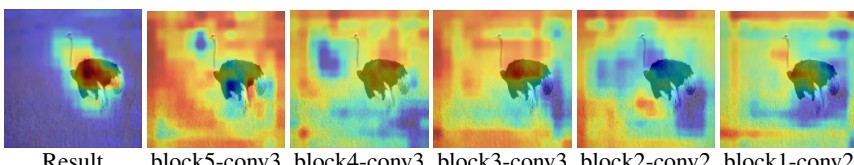

Result    block5-conv3    block4-conv3    block3-conv3    block2-conv2    block1-conv2

Figure 7: Sanity check results for MI-Grad-CAM on the VGG-16.

## 4.7 SANITY CHECK

As recommended in Adebayo et al. (2018), which highlights the importance of performing a sanity check on saliency maps beyond visual assessment alone, we perform a cascading parameter randomization test to evaluate the sanity of our method. Fig. 7 presents the sanity check results for MI-Grad-CAM, obtained by progressively randomizing the parameters from shallow to deeper layers. As shown, the saliency map deteriorates with each step of random initialization in the network, indicating its sensitivity to model parameters. This sensitivity suggests that MI-Grad-CAM effectively reflects the model's quality.

## 5 EXPERIMENTAL RESULTS ON VIT

We also evaluated the effectiveness of MI-Grad-CAM on ViT-B/16 Dosovitskiy et al. (2020) through qualitative and quantitative experiments. Results summarized in Table 4 indicate that MI-Grad-CAM performs better than all other baselines in explaining the ViT, scoring AD, AI, and HCI of 46.86%, 90.00%, and 0.668, respectively. MI-Grad-CAM also scores a win percentage of 56.6%. This sug-

Table 4: Evaluation results on ViT-B/16.

| Method | AD↓ | AI↑ | HCI↑ | Win%↑ |
|---|---|---|---|---|
| Grad CAM | 83.725 | 61.529 | 0.257 | 6.6 |
| Grad CAM++ | 56.045 | 86.564 | 0.583 | 6.6 |
| Score CAM | 69.797 | 64.637 | 0.412 | 13.3 |
| Eigen CAM | 59.742 | 55.671 | 0.467 | 16.6 |
| MI-Grad-CAM | **46.861** | **90.001** | **0.668** | **56.6** |

gests that MI-Grad-CAM can provide reliable explanations for transformer-based architectures apart from CNNs. MI-Grad-CAM generates visually coherent and high-fidelity heatmaps that distinctly emphasize the true special area of model attention while minimizing artifacts. The heatmap visualizations of MI-Grad-CAM and other CAM methods are provided in the Appendix A.6.

## 6 CONCLUSION

In this paper, we proposed MI-Grad-CAM, a novel post-hoc explanation method that enhances model interpretability by utilizing mutual information between the input and feature maps, along with gradients of the predicted class score, to determine the weighting for each feature map in the CAM. Our approach advances visual explanations by emphasizing causality, ensuring that highlighted regions closely correspond to model decision-making processes. Validated through both qualitative and quantitative experiments, MI-Grad-CAM demonstrates superior performance in faithfulness and localization metrics, consistently outperforming existing CAM methods. Counterfactual analysis confirms the causal validity of our explanations. Additionally, we introduced the HCI as a balanced metric to assess explanation quality, further establishing the robustness of MI-Grad-CAM.

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

# A   APPENDIX

## A.1   MATHEMATICAL PROOF OF OUR PROPOSAL

Let $X$ be an input image, and $A_k^l$ be the feature map of the $k^{th}$ neuron of the $l^{th}$ layer.

### OBJECTIVE

The feature maps can be weighted with the weights of $A_k^l$ being $w_k$ for generating a CAM with:

$$w_k = \alpha_k^c \cdot m_k,$$

where

$$\alpha_k^c = \frac{1}{Z} \sum_{ij} \frac{\partial Y^c}{\partial A_k^{l(i,j)}},$$

and $m_k$ is normalized $I(X; A_k^l)$:

$$m_k = \frac{I(X; A_k^l)}{\sum_k I(X; A_k^l)}.$$

Here $(i, j)$ denotes the spatial locations in the feature map $A_k^l$. The relationship between a feature map $A_k^l$ and the input image $X$ is analyzed in three cases:

### CASE 1: INFORMATIVE BUT NOT CLASS-SPECIFIC

$A_k^l$ is informative of $X$, but does not have a class-specific influence on the predicted class $c$.

$$w_k = \frac{1}{Z} \sum_{ij} \frac{\partial Y^c}{\partial A_k^{l(i,j)}} \cdot \frac{I(X; A_k^l)}{\sum_k I(X; A_k^l)}$$

Now, $Y^c$ of the neural network is a function of intermediate activations:

$$Y^c = f(A^1, A^2, \ldots, A^l, \ldots, A^N),$$

where $N$ is the total number of layers.

For a small perturbation $\epsilon$ on $A_k^l$:

$$A_k^{l(i,j)} \to A_k^{l(i,j)} + \epsilon.$$

The change in $Y^c$ due to this perturbation can be expressed as:

$$Y^c(A_k^{l(i,j)} + \epsilon) = Y^c(A_k^{l(i,j)}) + \epsilon \cdot \frac{\partial Y^c}{\partial A_k^{l(i,j)}} + O(\epsilon^2).$$

If $Y^c$ is not dependent on $A_k^l$:

$$Y^c(A_k^{l(i,j)} + \epsilon) = Y^c(A_k^{l(i,j)}) \quad \Rightarrow \quad \frac{\partial Y^c}{\partial A_k^{l(i,j)}} = 0.$$

Thus, the weight of feature map $A_k^l$ becomes:

$$w_k = \frac{1}{Z} \sum_{ij} \frac{\partial Y^c}{\partial A_k^{l(i,j)}} \cdot \frac{I(X; A_k^l)}{\sum_k I(X; A_k^l)} \approx 0.$$

Therefore the weight assigned to the feature map not being class influential is low.

## CASE 2: CLASS-SPECIFIC BUT NOT INFORMATIVE

$A_k^l$ is correlated to a class-specific prediction score $c$, but is not informative of the input $X$.

If feature map $A_k^l$ is independent of the input $X$ Then probability distribution of $X$ and $A_k^l$ are independent :

$$P(X, A_k^l) = P(X)P(A_k^l) \quad \text{and} \quad \frac{P(X, A_k^l)}{P(X)P(A_k^l)} = 1.$$

From the formula for mutual information:

$$I(X; A_k^l) = \sum_{X, A_k^l} P(X, A_k^l) \log \frac{P(X, A_k^l)}{P(X)P(A_k^l)} = 0.$$

Additionally:

$$I(X; A_k^l) = H(X) + H(A_k^l) - H(X, A_k^l) = H(X) - H(X|A_k^l),$$

where $H(X)$ is the entropy of $X$, $H(X, A_k^l)$ is the joint entropy, and $H(X|A_k^l)$ is the conditional entropy of $X$ given $A_k^l$.

If $I(X; A_k^l) = 0$:

$$H(X) - H(X|A_k^l) = 0 \quad \Rightarrow \quad H(X) = H(X|A_k^l).$$

This indicates there is no reduction in the randomness of $X$ even after knowing $A_k^l$, implies $A_k^l$ provides no information regarding $X$.

Thus, the weight of $A_k^l$:

$$w_k = \frac{1}{Z} \sum_{ij} \frac{\partial Y^c}{\partial A_k^{l(ij)}} \cdot \frac{I(X; A_k^l)}{\sum_k I(X; A_k^l)} \approx 0.$$

## CASE 3: INFORMATIVE AND CLASS-SPECIFIC

$A_k^l$ is informative of $X$ and also has a class-specific influence on the predicted class $c$.

If $A_k^l$ is informative of $X$:

$$H(X) > H(X|A_k^l) \quad \Rightarrow \quad I(X; A_k^l) > 0.$$

Since $Y^c$ is dependent on $A_k^l$:

$$\frac{\partial Y^c}{\partial A_k^{l(i,j)}} \neq 0.$$

Thus, the weight of feature map $A_k^l$:

$$w_k = \frac{1}{Z} \sum_{ij} \frac{\partial Y^c}{\partial A_k^{l(ij)}} \cdot \frac{I(X; A_k^l)}{\sum_k I(X; A_k^l)} > 0.$$

This implies $w_k > 0$ for feature map $A_k^l$, as it is correlated with $Y^c$ and also informative of $X$.

### A.2 HARMONIZED CONFIDENCE INDEX (HCI)

In XAI, evaluating explanation quality is crucial, particularly in sensitive domains like medical diagnostics. Commonly used metrics, such as *Average Increase (AI)* and *Average Drop (AD)*, assess different aspects of explanation fidelity. However, each metric individually may not fully capture the balance between maintaining model confidence and enhancing relevance to the target class. To address this, we propose a new metric, the *Harmonized Confidence Index (HCI)*, which combines *AI* and *AD* through a harmonic mean to deliver a balanced assessment of explanation quality.

***Definition: HCI :*** The *HCI* is defined as:

$$\text{HCI} = 2 \left( \frac{(1 - \text{AD}) \cdot \text{AI}}{(1 - \text{AD}) + \text{AI}} \right), \tag{14}$$

where:

- AI $\in [0, 1]$: Measuring the improvement in the model's confidence when focusing on explanation-relevant regions.

- AD $\in [0, 1]$: Quantifying the reduction in the model's confidence when relying solely on explanation-relevant features.

- $(1 - \text{AD}) \in [0, 1]$: Confidence retention, representing how much confidence is preserved when explanations are applied.

The HCI combines AI and $(1 - \text{AD})$ using a harmonic mean to balance confidence enhancement and retention.

### A.2.1 MOTIVATION FOR USING THE HARMONIC MEAN

The harmonic mean, defined as:

$$\text{HM}(a, b) = \frac{2ab}{a + b}, \tag{15}$$

is particularly suitable for combining metrics that must both be high to achieve a strong overall score. It penalizes low values, ensuring that if either *AI* or $(1 - \text{AD})$ is low, the HCI score will also be low. This behavior aligns with the requirements of XAI evaluation, where both confidence retention and relevance are essential.

### A.2.2 THEORETICAL PROOF OF HCI

***Lemma 1: Range and Interpretation of HCI***
***Statement:*** The HCI metric lies in the range $[0, 1]$.

***Proof:***

1. By definition, AI $\in [0, 1]$ and $(1 - \text{AD}) \in [0, 1]$. These values are bounded because they represent normalized confidence scores.

2. The harmonic mean of two non-negative variables $a$ and $b$ is defined in Eq. (15) satisfies the property:

$$\min(a, b) \leq \text{HM}(a, b) \leq \max(a, b). \tag{16}$$

3. Substituting $(1 - \text{AD})$ and AI as $a$ and $b$, we have:

$$\min((1 - \text{AD}), \text{AI}) \leq \text{HCI} \leq \max((1 - \text{AD}), \text{AI}). \tag{6}$$

4. Since both $(1 - \text{AD})$ and AI lie in $[0, 1]$, it follows that:

$$0 \leq \text{HCI} \leq 1. \tag{7}$$

Thus, HCI is bounded within the range $[0, 1]$.

***Lemma 2: Symmetry of HCI***
***Statement:*** The HCI metric is symmetric with respect to AI and $(1 - \text{AD})$.

***Proof:***

1. The HCI formula can be rewritten as:

$$\text{HCI} = \frac{2 \cdot \text{AI} \cdot (1 - \text{AD})}{\text{AI} + (1 - \text{AD})}. \tag{8}$$

2. By the commutative property of multiplication and addition, swapping AI and $(1 - AD)$ in the numerator and denominator does not change the result:

$$\text{HCI} = \frac{2 \cdot (1 - AD) \cdot AI}{(1 - AD) + AI}. \tag{9}$$

3. This symmetry ensures that both AI and $(1 - AD)$ contribute equally to the HCI score.

Thus, HCI is symmetric with respect to its components.

***Lemma 3: Sensitivity to Low Values***
***Statement:*** HCI penalizes low values of either AI or $(1 - AD)$.

***Proof:***

1. Considering HCI formulation in Eq. (9).

2. If $AI \rightarrow 0$, the numerator becomes zero:

$$\text{HCI} = \frac{2 \cdot 0 \cdot (1 - AD)}{0 + (1 - AD)} = 0. \tag{10}$$

3. Similarly, if $(1 - AD) \rightarrow 0$, the numerator becomes zero:

$$\text{HCI} = \frac{2 \cdot AI \cdot 0}{AI + 0} = 0. \tag{11}$$

This behavior demonstrates that HCI decreases sharply if either AI or $(1 - AD)$ is low. Thus, HCI penalizes explanations that fail to achieve either relevance (AI) or retention $((1 - AD))$.

***Theorem : HCI as a Balanced Metric***
***Statement:*** HCI provides a balanced evaluation of confidence retention $(1 - AD)$ and confidence enhancement (AI).

***Proof:***

1. **Balanced Contribution:** The harmonic mean ensures that AI and $(1 - AD)$ contribute equally to HCI in Eq. (9).

2. **Penalization of Imbalance:** If one component is significantly smaller than the other, HCI decreases, penalizing explanations that fail in either relevance or retention.

3. **Best and Worst Cases:**
   **Best Case:** When $AI = 1$ and $AD = 0$, the Eq. (9) becomes:

$$\text{HCI} = 2 \cdot \frac{1 \cdot 1}{1 + 1} = 1, \tag{12}$$

   indicating perfect explanation quality.
   **Worst Case:** When $AI = 0$ or $(1 - AD) = 0$, the Eq. (9) becomes:

$$\text{HCI} = 0, \tag{13}$$

   indicating a complete failure in either confidence enhancement or retention.

Thus, HCI balances confidence retention and enhancement effectively.

### A.2.3    ADVANTAGES OF HCI

The HCI metric offers the following benefits:

1. HCI combines *AI* and $(1 - AD)$ to ensure high performance in both confidence retention and enhancement.

2. HCI penalizes low scores for either component, encouraging improvement in both metrics.

3. HCI provides a unified score between 0 and 1, making it intuitive and easy to interpret.

### A.2.4 Application in Explainable AI Evaluation

The HCI is particularly suited for evaluating Class Activation Mapping (CAM) and other heatmap-based explanation methods. By utilizing both confidence retention and relevance, HCI can provide a more comprehensive understanding of an explanation's quality, thereby aiding researchers and practitioners in developing and selecting more trustworthy XAI models. This metric could be significant in domains requiring high reliability, such as medical diagnostics, where both maintaining and enhancing confidence are essential to explanation quality.

### A.3 Limitations in Existing CAM Methodologies

1. CAMs guide activation maps toward regions significant for a specific class, which can bias attention and overlook other relevant areas. As a result, they often yield output-centric rather than input-centric, information-rich visualizations (Fig. 8).

2. Existing CAMs ignore direct relationships between feature maps and the input image, focusing only on target scores. This limits their ability to reveal the model's focus, attended features, and spatial cues for unseen or random classes.

3. CAMs mainly capture correlation, not causation. While they highlight influential regions for a class, they do not quantify how much information a feature provides about the input.

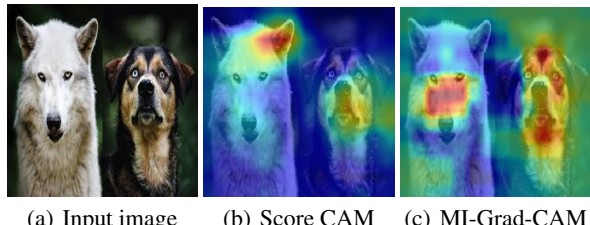

(a) Input image     (b) Score CAM     (c) MI-Grad-CAM

Figure 8: Visualizations generated by Score-CAM (b) and our proposed MI-Grad-CAM (c). In (c), the model focuses on the nasal and central facial regions of both animals for decision-making, which are not highlighted in Score-CAM (b).

### A.4 Quantitative Analysis Results on Cats and Dogs Dataset

Table 5 presents the results of this analysis carried out on the ResNet-50 and VGG-16 pre-trained models on AD, AI, HCI and win percentage metrics.

Table 5: Evaluation results for AD, AI, HCI, and win percentage on Cats and Dogs validation dataset. Lower AD, higher AI, higher HCI, and higher win percentage indicate better performance of the explainability method.

| Th. | M | Method | AD ↓ | AI ↑ | HCI ↑ | Win ↑ |
|-----|---|--------|------|------|-------|-------|
| 0.5 | VGG-16 | Grad CAM | 92.132 | 43.700 | 0.133 | 20.0 |
| | | Grad CAM++ | 98.740 | 30.869 | 0.024 | 6.6 |
| | | Score CAM | 89.354 | 45.896 | 0.175 | 23.3 |
| | | XGrad CAM | 93.860 | 42.163 | 0.107 | 13.3 |
| | | Eigen CAM | 99.927 | 13.151 | 0.001 | 6.6 |
| | | MI-Grad-CAM | **86.655** | **48.029** | **0.208** | **30.0** |
| | ResNet-50 | Grad CAM | 87.079 | 70.300 | 0.219 | 16.6 |
| | | Grad CAM++ | 93.579 | 55.305 | 0.115 | 6.6 |
| | | Score CAM | **71.211** | 30.884 | **0.297** | 26.6 |
| | | XGrad CAM | 98.869 | 39.722 | 0.022 | 10.0 |
| | | Eigen CAM | 97.490 | 29.844 | 0.046 | 6.6 |
| | | MI-Grad-CAM | 83.160 | **76.317** | 0.276 | **33.3** |

From the statistics it can be inferred that MI-Grad-CAM achieves AD of 86.65% and 83.316% and AI of 48.02% and 76.31% in VGG-16 and ResNet-50 respectively. MI-Grad-CAM also achieves

HCI scores of 0.208 and 0.276 across these architectures, outperforming all other gradient-based CAM methods. Similarly, MI-Grad-CAM showcases superior performance in terms of the win percentage metric by achieving win percentages of 30.0% and 33.3% on VGG-16 and ResNet-50 respectively. This highlights the generalizability of MI-Grad-CAM to perform across datasets and different model architectures.

## A.5 INTERMEDIATE LAYER RESULTS

This section presents the quantitative analysis of MI-Grad-CAM using AD, AI, and HCI metrics on the intermediate layers of ResNet-50 and DenseNet-121 architectures.

Table 6: Evaluation results for AD, AI and HCI for intermediate layer of ResNet-50. Lower AD, Higher AI and Higher HCI signifies a better performance of the explainability method.

| Method | Avg. Drop (AD) ↓ | Avg. Increase (AI) ↑ | HCI ↑ |
|---|---|---|---|
| Grad CAM | 89.4284 | 60.2767 | 0.180 |
| Grad CAM++ | 94.3329 | 36.6570 | 0.098 |
| Score CAM | **57.4353** | 37.0429 | **0.396** |
| XGrad CAM | 95.0017 | 34.2830 | 0.087 |
| Eigen CAM | 99.6508 | 18.6089 | 0.007 |
| MI-Grad-CAM | 82.4394 | **71.2767** | 0.282 |

Table 7: Evaluation results for AD, AI and HCI for intermediate layer of DenseNet-121. Lower AD, Higher AI and Higher HCI signifies a better performance of the explainability method.

| Method | Avg. Drop (AD) ↓ | Avg. Increase (AI) ↑ | HCI ↑ |
|---|---|---|---|
| Grad CAM | 69.4763 | 64.5142 | 0.414 |
| Grad CAM++ | **65.3527** | 50.6910 | 0.421 |
| Score CAM | 74.4789 | 62.0164 | 0.362 |
| XGrad CAM | 74.0283 | 57.1403 | 0.357 |
| Eigen CAM | 79.4558 | 40.3209 | 0.272 |
| MI-Grad-CAM | 66.4773 | **68.8142** | **0.451** |

The results in Table 6 and Table 7 indicate that MI-Grad-CAM outperforms other gradient-based CAM methods on intermediate layers, achieving HCI scores of 0.282 and 0.451 for ResNet-50 and DenseNet-121, respectively. These findings highlight the superior performance of MI-Grad-CAM in capturing relevant features across different layers.

## A.6 QUALITATIVE ANALYSIS RESULTS USING VISION TRANSFORMER (VIT)

Fig. 9 presents the heatmap visualizations of our proposed approach, MI-Grad-CAM and other CAM methods on ViT-B/16. Our approach generates visually coherent and high-fidelity heatmaps that distinctly emphasize the true spatial areas of model attention while effectively minimizing artifacts.

## A.7 EMPIRICAL INFERENCE TIME ANALYSIS

One major draw back often associated with the use of mutual information in high dimensional data analysis is the high time consumption or high inference runtime that takes place during the mutual information calculation. Hence, we analyze and compare the compute time consumed by state-of-the-art CAM methods with our proposed approach.

Table 8 summarizes our inference time analysis results in three devices, CPU, T4 GPU and TPU v2-8, for MI-Grad-CAM and rest CAM approaches on VGG-16 and ResNet-50. Each result is the average taken over 10 runs on the sample. It can be seen that though better performance comes at higher computational cost, MI-Grad-CAM performs faster than Score CAM with around $2\times$ and $27\times$ less runtime on T4 GPU and CPU respectively while providing comparable performance on VGG-16.

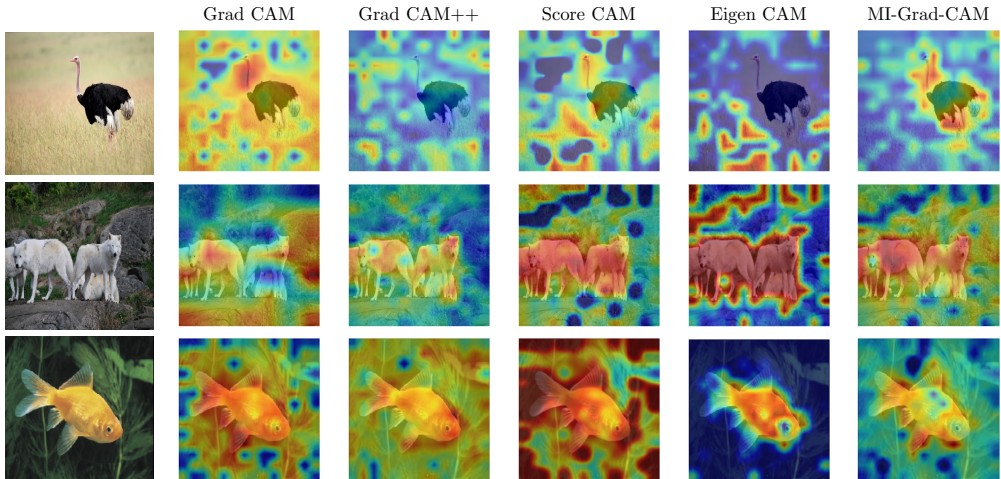

Figure 9: Visualization results of state-of-the-art CAM methods: Grad-CAM, Grad-CAM++, Score-CAM, Eigen-CAM and our proposed MI-Grad-CAM on ViT-B/16.

Table 8: Empirical Inference Time (s) Analysis on VGG-16 (M1) and ResNet-50 (M2).

| M | Device | Grad-CAM | Grad-CAM++ | ScoreCAM | EigenCAM | MI-Grad-CAM |
|---|---|---|---|---|---|---|
| | CPU | 4.02 | 12.21 | 386.85 | 5.07 | 13.88 |
| M1 | T4 GPU | 0.59 | 0.33 | 13.58 | 0.94 | 6.53 |
| | TPU v2-8 | 3.66 | 10.39 | 43.90 | 4.93 | 5.19 |
| | CPU | 15.63 | 13.30 | 556.34 | 10.62 | 33.88 |
| M2 | T4 GPU | 4.56 | 1.15 | 240.95 | 4.68 | 18.83 |
| | TPU v2-8 | 4.39 | 8.13 | 360.07 | 2.01 | 17.99 |

## A.8 ABLATION STUDY

To demonstrate the effectiveness of including mutual information in the weight score calculation of feature maps in CAM, we conducted weighting component visualizations and counterfactual comparison analyses. This section presents the ablation study results, highlighting the effectiveness of incorporating mutual information into the proposed framework. These results underscore the contribution of mutual information to the overall performance, demonstrating its impact within our model's design.

### A.8.1 WEIGHTING COMPONENT VISUALIZATION

For a weighting criterion to be valid in CAM, it must ensure stability, avoid noise, and remain robust under different conditions. To justify the inclusion of mutual information in the weighting process, we visualize the weighting components with and without mutual information across the channels.

Fig. 10 compares the weighting components in both scenarios. Without mutual information, the weights are noisy and unstable across all instances, undermining their reliability. However, when mutual information is integrated, the weights demonstrate significant robustness, effectively addressing the noisy weight problem and maintaining stability.

### A.8.2 COUNTERFACTUAL COMPARISON ANALYSIS

As highlighted in the paper, counterfactual analysis evaluates the causal validity of explanations generated by MI-Grad-CAM. In this section, we provide comparative plots before and after incorporating mutual information weight components to further validate its potency in generating causal explanations (Fig. 11). The results demonstrate that explanations without mutual information weight components lack sensitivity to counterfactual perturbations, failing to capture causality. In contrast, incorporating mutual information significantly enhances the model's sensitivity to counterfactual

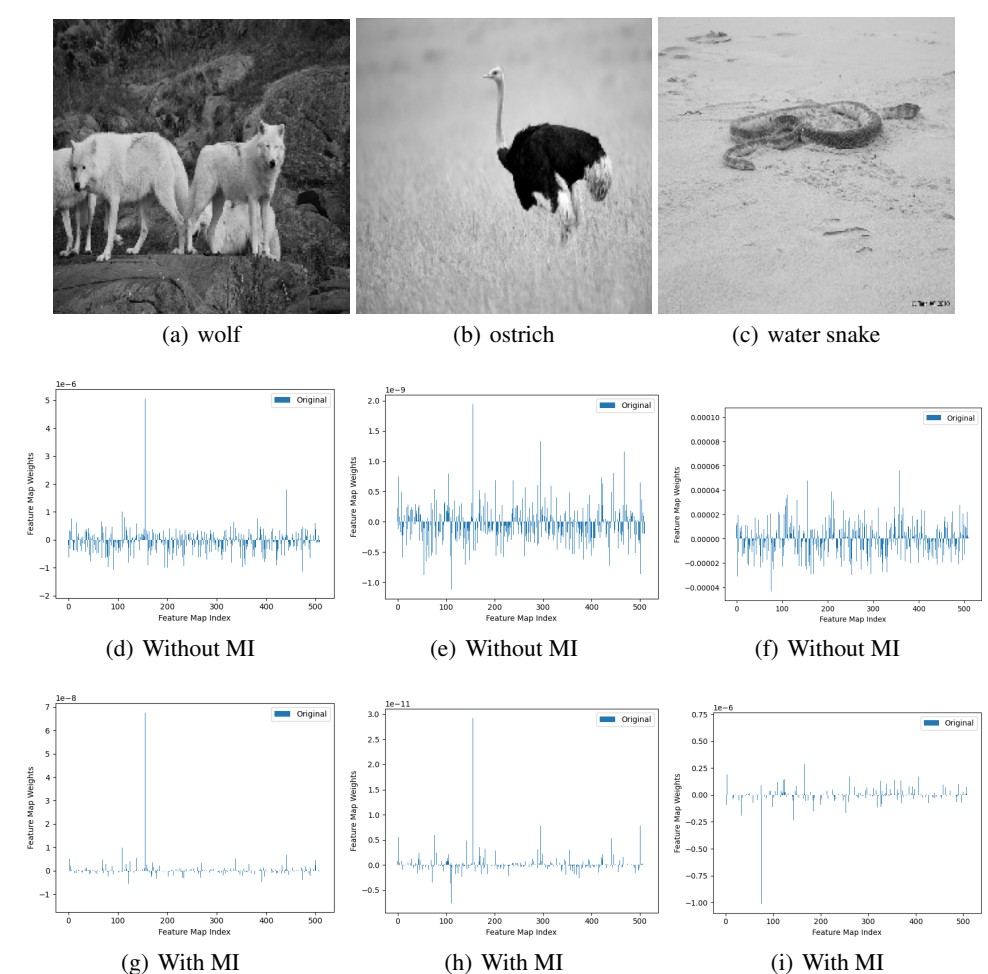

Figure 10: Weight component visualizations for three instances from the ILSVRC2012 dataset across channels. The inclusion of mutual information leads to more stable and less noisy weights per feature maps, demonstrating its effectiveness.

changes, producing explanations that are more causal. These findings reaffirm the rationale for integrating mutual information into the weighting criteria of feature maps, as discussed earlier.

### A.8.3 SENSITIVITY ANALYSIS TO DIFFERENT ACTIVATION FUNCTIONS

MI-Grad-CAM incorporates a ReLU operation on the activation map produced for finer visualization, as defined in Eq. (18), Definition 4 of the paper. In this section, we analyze the sensitivity of our approach to different activation functions and demonstrate that ReLU results in the best quantitative results in comparison to other activation functions. Results corresponding to the comparative analysis are summarized in Table 9. It can be inferred from the quantitative results that while ReLU and ELU provide consistently strong results with HCI values of 0.29, Sigmoid remains competitive as well-especially in HCI (0.25). Leaky ReLU, however, shows some degradation in the reliability metrics. Hence, we adopt the ReLU activation function in our approach.

### A.8.4 IMPORTANCE OF GRAYSCALING AND UPSAMPLING

Grayscaling reduces the input's spectral dimensionality, ensuring compatibility with the single-channel structure of the activation maps. This step simplifies pairwise comparisons, as mutual information requires a pairwise correspondence between elements in the activation map and the input image. Additionally, maintaining a consistent number of samples in both variables, $A_l^k$ and

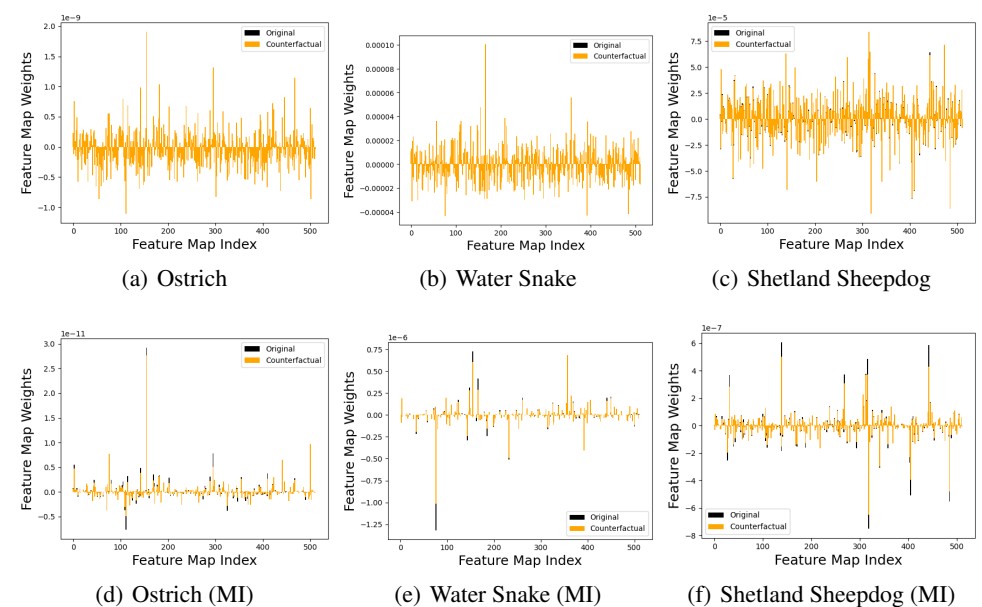

(a) Ostrich       (b) Water Snake       (c) Shetland Sheepdog

(d) Ostrich (MI)       (e) Water Snake (MI)       (f) Shetland Sheepdog (MI)

Figure 11: Comparative counterfactual plots for three instances from the ILSVRC2012 dataset across channels. Plots without mutual information exhibit minimal to no sensitivity to counterfactual changes, emphasizing the significance.

Table 9: Sensitivity Analysis of MI-Grad-CAM to Non-ReLU Activation Functions.

| Metric | ReLU | Leaky ReLU | Sigmoid | ELU |
|---|---|---|---|---|
| Avg. Drop (AD) $\downarrow$ | 79.32 | 89.52 | 81.32 | 79.33 |
| Avg. Increase (AI) $\uparrow$ | 46.21 | 26.85 | 42.20 | 46.20 |
| HCI $\uparrow$ | 0.29 | 0.15 | 0.25 | 0.29 |

$X$, is necessary for joint probability calculation, which requires vectors of identical lengths. This underscores the importance of upsampling and the grayscale transformation.

### A.8.5 MI-GRAD-CAM VISUALIZATIONS ON GRAYSCALE IMAGES

MI-Grad-CAM, as defined in Sec.3.1.1, leverages a grayscale transformation on the input images prior to the mutual information calculation. This is adopted to align the spectral dimensionality of the input images with that of the activation maps, crucial for mutual information calculation. Moreover, this enables to reduce the computational overhead of our approach, making it scalable. This operation can be presumed to result in loss of discriminative information when foreground and background regions have similar grayscale intensities. Hence, in our experiments, we empirically evaluated the impact of this transformation by comparing MI-Grad-CAM maps with and without color information. Visualizations corresponding the same are presented in Fig. 12. The results showed negligible degradation in localization performance, indicating that the grayscale transformation was sufficient to capture the salient regions relevant to the task.

### A.9 CAUSALITY ANALYSIS IN SCORE CAM

In accordance with the claim that we raise in our work, i.e the lack of guarantee and analysis of causality in explanations generated by earlier CAM approaches, we report the counterfactual analysis plots of Score CAM.

It can be inferred from Fig 13 that Score CAM shows no sensitivity to counterfactual changes and hence, no causality in its explanations. This emphasizes the importance asserted on the causality of explanations in our study and the MI-Grad-CAM approach.

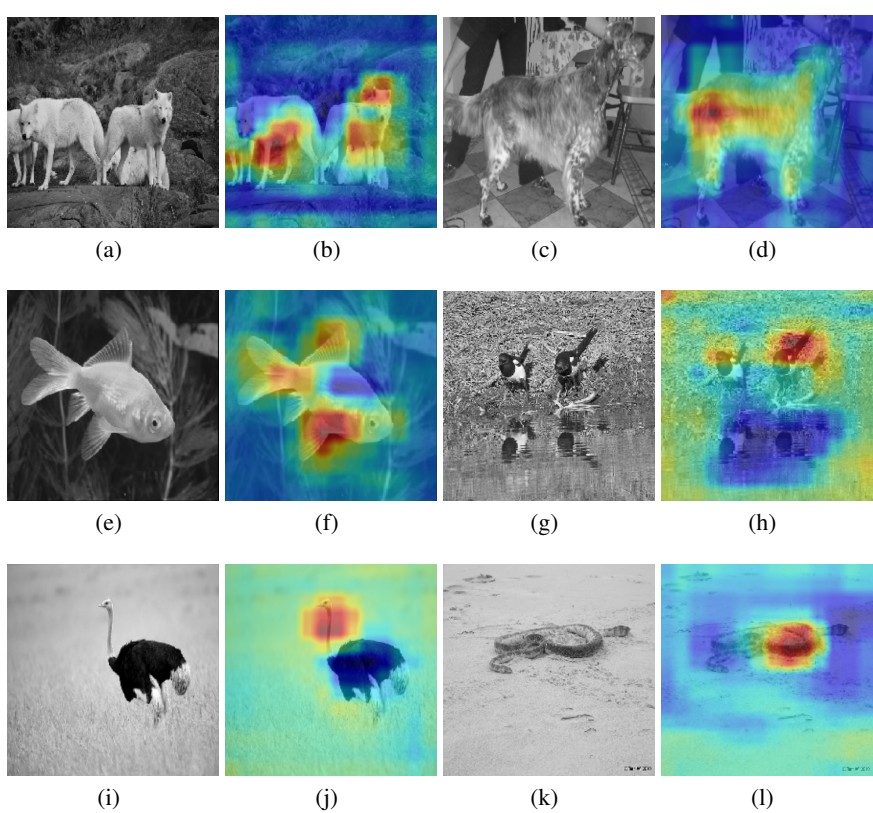

Figure 12: Visualizations of MI-Grad-CAM on grayscale images. Minimal to no loss in localization indicates grayscale transformation doesn't degrade CAM quality.

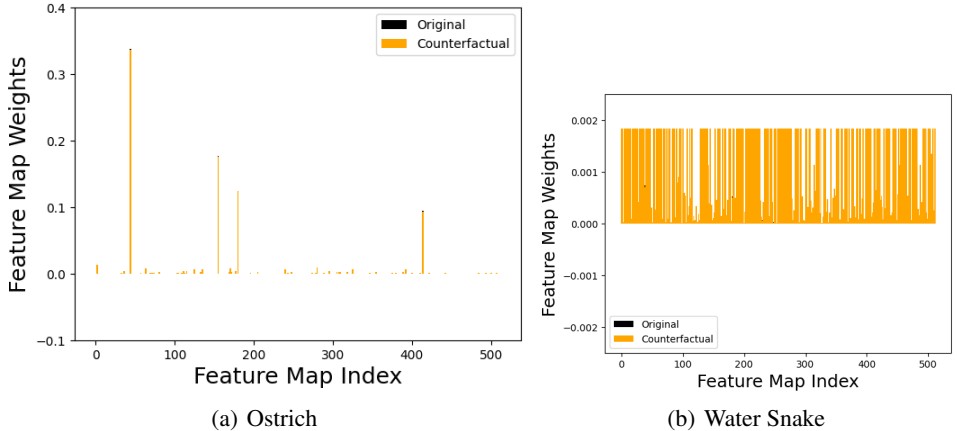

(a) Ostrich                                    (b) Water Snake

Figure 13: Counterfactual plots for two instances from the ILSVRC2012 dataset across the channels for Score CAM. Plots show feeble to no signs of causality.

## A.10 ADDITIONAL VISUALIZATIONS

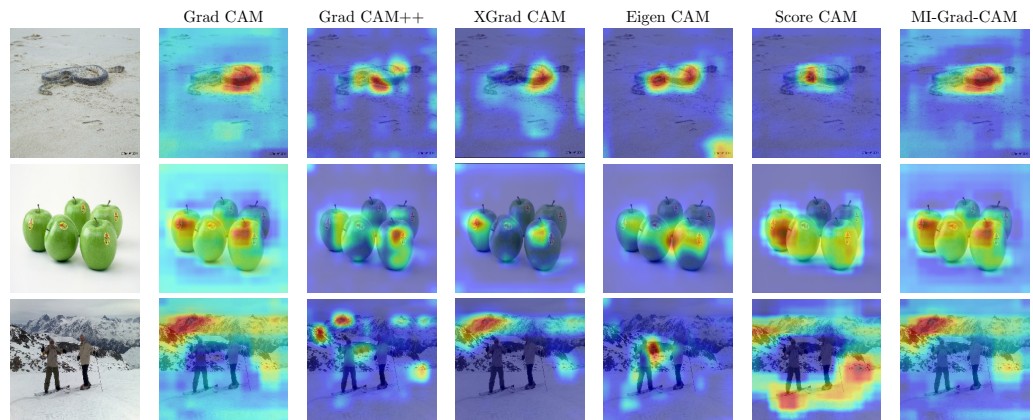

Figure 14: Additional visualizations on the ILSVRC2012 validation dataset. The target classes are 'water snake', 'granny smith' and 'alp' from top to bottom.

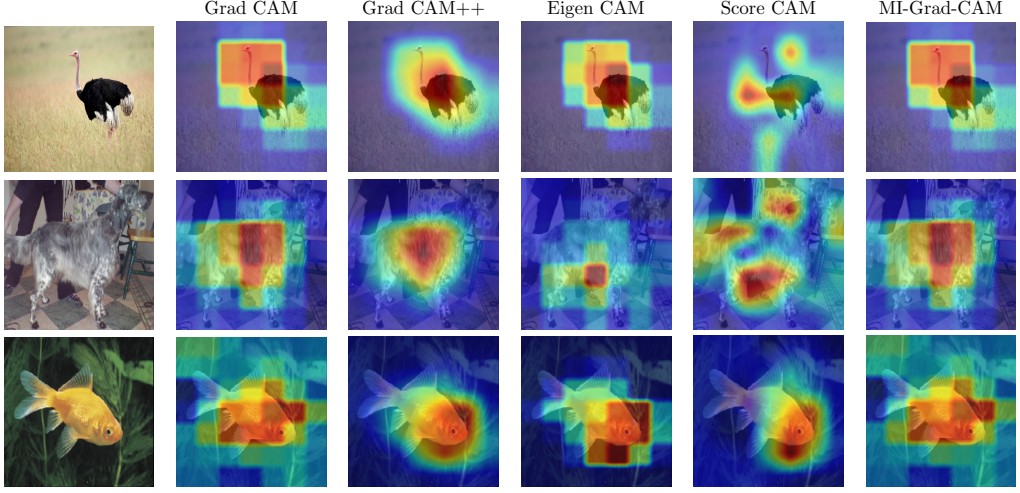

Figure 15: Comparative visualizations of state-of-the-art CAM methods Grad-CAM, Grad-CAM++, Score-CAM, Eigen-CAM and our proposed approach MI-Grad-CAM on ResNet-50.

