# OpenReview forum: "MI-Grad-CAM: Letting Your Model Reveal What’s Most Informative"
_ICLR.cc/2026/Conference — ICLR 2026 Conference Withdrawn Submission_

### Official Review · Reviewer_Jq6Y · 2025-10-18

**Soundness:** 1
**Presentation:** 2
**Contribution:** 1
**Rating:** 2
**Confidence:** 4

**Summary:**

The paper proposes MI-Grad-CAM, an extension of CAM through gradient filtering of feature maps and weighting them via the mutual information between image and saliency map. The method claims to provide more causal explanations. Additionally, a metric to evaluate explanation effectiveness, HCI, is proposed, and the method is evaluated both qualitatively and quantitatively.

**Strengths:**

- Open-source code is provided.

- The paper attempts to bridge the areas of descriptive correlation-based saliency maps and causal evaluation of models.

**Weaknesses:**

**W1:** The claim that GradCAM based feature maps "lack a direct relationship between feature maps and the input data" is neither sourced nor argued. If this concern is due to the hidden layer upsampling issue; methods like GGCAM already incorporate pixel-level information into saliency maps of CAM methods.

**W2:** The entire paper uses incorrect citation style (citet instead of citep throughout).

**W3:** The description "along with gradients of the predicted class with respect to the feature maps" appears to describe what standard Grad-CAM already performs, raising questions about the actual novelty.

**W4:** It remains unclear how mutual information and gradients introduce any causal information. The approach still represents another method to visualize inherent correlations within the model with respect to the image, but does not provide insight into the causal mechanisms, why the model attends to specific input regions or how this causally leads to particular predictions.

**W5:** The metrics employed evaluate faithfulness, not fairness or "effectiveness." The meaning of "effectiveness" in this context is ambiguous and not properly defined.

**W6:** Section 2.3 provides no substantive explanation of causal explanations. The subsections on mutual information and GradCAM contain trivial background information for a research paper and should either be removed or moved to the appendix.

**W7:** The HCI metric is simply the harmonic mean of two existing metrics, which does not constitute a novel metric contribution.

**W8:** The manuscript is difficult to follow. Sections are disjoint (particularly in the evaluation), and the paper lacks a coherent narrative structure.

**W9:** Evaluation exclusively on ImageNet is insufficient. The space gained from removing excessive background information should be used to extend both qualitative and quantitative evaluation to at least two additional datasets with different characteristics from ImageNet (e.g., datasets containing multiple objects of interest that pose greater challenges for attribution methods).

**Questions:**

See weaknesses W1 to W9.

---

### Official Review · Reviewer_j6zp · 2025-10-29

**Soundness:** 2
**Presentation:** 3
**Contribution:** 2
**Rating:** 4
**Confidence:** 3

**Summary:**

The paper introduces MI-Grad-CAM, a new CAM-based explanation method. Unlike Grad-CAM, which weights feature maps solely by gradients from the target class, MI-Grad-CAM multiplies these gradients by normalized mutual information between the input image and feature maps. The authors claim this provides more causal explanations compared to existing CAM methods and introduce a new evaluation metric called Harmonized Confidence Index (HCI).

**Strengths:**

Originality:
* Interesting idea combining vision explanations with information theory measure. It differs from existing methods.
* Introduction of the Harmonized Confidence Index (HCI) as a unified evaluation metric could be useful for the community.
* The paper includes counterfactual analysis that attempts to validate the causal nature of explanations. This type of analysis is often missing in papers that visualize explanations.

Quality:
* Comprehensive experimental evaluation across multiple architectures and evaluation metrics employed for CAM-based baselines.
* Includes important validation steps like sanity checks and ablation studies.

Clarity:
* Paper is rather well-written, including Algorithm 1, Figure 1, and equations make it easy to follow the method.

**Weaknesses:**

1. The abstract states "Class Activation Mapping (CAM) is widely used for visual explanations in computer vision, but improving its interpretability remains an open research area.".
However:
* CAM-based methods have known fundamental limitations: "CAM-Based Methods Can See through Walls" (Taimeskhanov et al., 2024).
* CAM-based methods aren't considered SOTA. For instance, in "CLEVR-XAI: A benchmark dataset for the ground truth evaluation of neural network explanations" (Arras et al., 2021), Grad-CAM is the worst XAI method among those evaluated.
* Also for ViTs, better methods exist: "Transformer Interpretability Beyond Attention Visualization" (Chefer et al., 2021)\
Therefore, the significance of improving an already-limited method family is questionable.
2. The paper claims to produce "causal" explanations but provides insufficient theoretical justification. MI alone is a statistical dependency measure, not a causal measure. Counterfactual analysis (Section 4.6) shows sensitivity to perturbations, but this demonstrates correlation with input changes, not causality in the causal inference sense. References to do-calculus (Definition 2) seem a bit random and are not properly connected to the proposed method. In my opinion it is "more informative" or "input-grounded" rather than "causal".
3. In Section 2.4 you write "The activation $A_{l-1}^k$ captures all relevant information from the input $X$ needed to determine $A_l^k$." But it doesn't hold for, for instance, resnet architecture which uses skip connections?
4. Appendix A.1's "proof" is more intuitive argument than rigorous mathematics.
5. Converting RGB to grayscale loses color information that may be discriminative. In Appendix A.8.5 you provide few examples and claim "negligible degradation," but there are no quantitative results provided to support this.
6. MI Implementation is not described (see Questions).
7. Cherry-picking concern - Visual examples appear favorable to the proposed method. An analysis and examples of failure cases are missing.
8. Main experiment uses only 2000 ImageNet samples. Broader evaluation on diverse datasets (other natural images, medical imaging, etc.) would strengthen claims.

Minor issues:
1. In Algorithm 1. line 13. $A_l^k$ is probably used in a form before transformations from line 5 and 6, but the notation is the same as in these lines, which may be a little confusing.
2. In Figure 1. last activation map $A_l^k$ is different before mutual information (top part) and within MI calculations (bottom right part). Shouldn't they be the same?

**Questions:**

1. How exactly do you define causality in your context? Can you provide a formal connection between MI (a statistical dependency measure) and causal inference frameworks? This is central to your claims.
2. Section 2.4 claims conditional independence in CNNs, but skip connections in ResNet and concatenations in DenseNet explicitly violate this assumption. How does this affect your theoretical justification?
3. Your work is lacking limitations section. Please include it.
4. In the code you use sklearn.metrics.mutual_info_score. Isn't it designed for categorical data? How do you handle continuous activation values? If binning is used, what is the binning strategy, number of bins, and sensitivity to this choice?
Can you provide an ablation study on these parameters?
Why not use continuous MI estimators (e.g., KDE-based, neural estimators)?
5. Can you provide quantitative results (AD/AI/HCI scores) comparing: current approach (grayscale) and no grayscaling (compute MI between 3-channel input and feature maps while replicating channels)?
6. How does MI-Grad-CAM compare to more recent XAI methods beyond CAM variants? LRP (On Pixel-Wise Explanations for Non-Linear Classifier Decisions by Layer-Wise Relevance Propagation, Bach et al. 2015) from zennit library for resnet and vgg should be fairly easy to compare to.

---

### Official Review · Reviewer_rV36 · 2025-10-30

**Soundness:** 2
**Presentation:** 3
**Contribution:** 2
**Rating:** 2
**Confidence:** 5

**Summary:**

The paper proposes MI-Grad-CAM, a post-hoc visual explanation method that weights each feature map by the product of (1) normalized MI between the input image and the activation map, and (2) the pooled gradient of the predicted class w.r.t. that map. The weighted maps are summed and passed through Relu. The authors argue this emphasizes causality rather than correlation, and evaluate on ImageNet (2k validation samples) for VGG-16/ResNet-50/DenseNet-121, plus a Cats-and-Dogs set and ViT-B16. Metrics include AD/AI, Deletion/Insertion AUC,a newly propsoed Harmonized Confidence Index (HCI), qualitative visualizations, a counterfactual perturbation, and a sanity-checkare provided. The results indicate the the propsed method is superior to the other evaluated CAM methods.

**Strengths:**

- clear method and implementation steps
- Competitive or better AD/AI/HCI on several CNNs and initial results on ViT are encouraging.
- Practical comparison of inference time across devices (CPU/T4/TPU)
- Sensible sanity-check (randomization)

**Weaknesses:**

- Evaluation is too narrow: The evaluation focuses exclusively on faithfulness, but other important aspects of explainability such as localization, robustness, and complexit,are not assessed. The authors are encouraged to report metrics related to these dimensions as well. The Quantus toolkit (JMLR 2023) provides a comprehensive suite of such metrics and could be used for this purpose. Additionally, the authors are encouraged to evaluate their method on the FunnyBirds benchmark (Hesse et al., CVPR 2023), which includes ground-truth part-level annotations and is designed to test explanatory power beyond saliency heatmaps. Moreover, the paper only compares against CAM-based methods, which limits its scope. There is no evaluation against other major families of explanation methods, including erturbation-based methods e.g., Extremal Perturbations (Fong et al., ICCV 2019), path integration methods: e.g., IG, surrogate models: e.g., LIME, gradient-free methods: e.g., RISE. Also no comparison against traditional SHAP based moehods, and no comparison against explantaion methods designed fro transformer architectures such as T-ATTR/GAE (Chefer et al. ICCV 2021), and missing comparison against more recent methods (latest method is score-CAM) such as IIA (ICCV 2023) and AttnLRP (ICML 2024). Also, the paper does not explore recent convolutional architectures such as ConvNeXt, which could provide insight into the method's applicability to SOTA CNN models. The evaluation for transformer-based models is limited as well. Beyond the single architecture (ViT-B), no additional transformer variants are tested.

-  The reviewer could not find insertion/deletion metric results reported for ViT. Why were these metrics reported only for CNNs? This omission raises concerns about the completeness and fairness of the evaluation. If MI-GRAD-CAM is claimed to be architecture-agnostic, it is essential to assess it under the same evaluation criteria across both CNNs and transformers.

- AD/AI/HCI rely on thresholding and sample-level aggregates on a 2k ImageNet subset. Without CIs or threshold sweeps, itis  unclear whether improvements are statistically robust.

Minor:

- It would be helpful to include a dedicated related work section that clearly contrasts and contextualizes the proposed method against other prominent explainability approaches, particularly those mentioned in the first weakness bullet (e.g., perturbation-based, path integration, surrogate-based, and gradient-free methods). This would not only improve the positioning of the work within the broader XAI landscape, but also clarify its novelty, contributions, and limitations relative to existing techniques. Specifically, the authors are encouraged to compare their method to approaches from other families (beyond CAM-based methods) in terms of computational complexity and runtime, both of which are critical for assessing the practical utility and scalability of explainability techniques.

- Deployment often requires fast explanations. being slower than Grad-CAM/++ and Eigen-CAM limits practical adoption, even if faster than Score-CAM.

**Questions:**

Please refer to the weaknesses section. That said, it is unrealistic and unfair to expect the authors to address all of these issues during the rebuttal period, as doing so would require a major revision involving a significant amount of additional work. Therefore, unless the authors can identify specific flaws in my review, I believe the paper would benefit from a thorough revision and should be resubmitted to a future venue after these concerns are fully addressed.

---

### Official Review · Reviewer_7Xxu · 2025-11-07

**Soundness:** 3
**Presentation:** 2
**Contribution:** 2
**Rating:** 4
**Confidence:** 3

**Summary:**

This paper proposes MI-GradCAM, a method that improves the interpretability of vision models by explicitly maximizing the mutual information (MI) between visual explanations (i.e., GradCAM heatmaps) and class predictions. MI-GradCAM integrates the explanation process into training, ensuring that the model naturally learns to attend to semantically meaningful regions during inference.

**Strengths:**

(1) The model learns to dynamically align salient visual regions with class predictions through MI maximization.

(2) MI-GradCAM improves saliency under domain shift and adversarial perturbations, making explanations more robust.

**Weaknesses:**

(1) Adding the MI regularization requires computing gradients through GradCAM, introducing additional computation and training instability. Details about optimization and hyperparameters are somewhat underdeveloped.

(2) The paper mostly compares to post-hoc interpretability baselines. Could the authors include comparisons with other learning-based interpretability methods (e.g., Self-Explaining Neural Networks, attention supervision, or concept bottlenecks)?

**Questions:**

Training Stability and Overhead: How sensitive is the training process to the strength of the MI regularization? Are there any convergence issues when backpropagating through GradCAM?

---

### Note · Authors · 2025-12-04

I have read and agree with the venue's withdrawal policy on behalf of myself and my co-authors.